# Sample Importance in Training Deep Neural Networks

**Tianxiang Gao, Vladimir Jojic**
Department of Computer Science
University of North Carolina at Chapel Hill
Chapel Hill, NC, 27599 , USA
`{tgao,vjojic}@cs.unc.edu`

## Abstract

The contribution of each sample during model training varies across training iterations and the model's parameters. We define the concept of sample importance as the change in parameters induced by a sample. In this paper, we explored the sample importance in training deep neural networks using stochastic gradient descent. We found that "easy" samples – samples that are correctly and confidently classified at the end of the training – shape parameters closer to the output, while the "hard" samples impact parameters closer to the input to the network. Further, "easy" samples are relevant in the early training stages, and "hard" in the late training stage. Further, we show that constructing batches which contain samples of comparable difficulties tends to be a poor strategy compared to maintaining a mix of both hard and easy samples in all of the batches. Interestingly, this contradicts some of the results on curriculum learning which suggest that ordering training examples in terms of difficulty can lead to better performance.

## 1 Introduction

Sample importance is the sample's contribution to the parameter change during training. In statistics, the concept "leverage" of a point is used (St Laurent & Cook (1992)) to measure the impact of a sample on the training of a model. In the context of SVM, the most important samples are the support vectors as they define the separating hyperplane. Understanding the importance of the samples can help us interpret trained models and structure training to speed up convergence and improve prediction accuracy. For example, Curriculum learning (CL) from Bengio et al. (2009) shows that training with easy samples first, then gradually transitioning to difficult samples can improve the learning. In CL, the "easiness" of a sample is predefined either manually or using an evaluation model. Self-paced learning (SPL) (Kumar et al. (2010)) shows that it is possible to learn from samples in order of easiness. In this framework, easiness is related to the prediction error and can be estimated from the model. However, easiness of a sample may not be sufficient to decide when it should be introduced to a learner. Maintaining diversity among the training samples can have a substantial effect on the training (Jiang et al. (2014)).

In this work, we explore the sample importance in deep neural networks. Deep learning methods have been successfully applied in many tasks and routinely achieve better generalization error than classical shallow methods (LeCun et al. (2015)). One of the key characteristics of a deep network is its capacity to construct progressively more complex features throughout its layers (Lee et al. (2011)). An intuitive question arises: which samples contribute the most to the training of the different layer's parameters? From literature Saxe et al. (2011), we know that even randomly generated filters can compute features that lead to good performance – presumably on easy samples. However, to learn hard samples correctly, the model may need to construct complex features, which require both more training time and refined filters from bottom layers. Hence, we hypothesized that the hard samples shape the bottom layers – closer to the input – and easy samples shape the top layers – closer to the output.

Motivated by the above hypothesis, we analyzed the sample importance in a 3 layer ReLU network on two standard datasets. The results reveal several interesting facts about the sample importance in easy and hard samples:

**1. Easy and hard samples impact the parameters in different training stages.** The biggest impact of easy samples on parameters are mostly during the early training stage, while the impact of hard samples become large in the late training stage.

**2. Easy and hard samples impact the parameters in different layers.** Easy samples impact have larger impact on top layer parameters, while hard samples shape the bottom layer parameters.

**3. Mixing hard samples with easy samples in each batch helps training.** We conducted batches with homogeneous or mixed "easiness". We found that use of homogeneous batches hinders the training. Hence, it is preferable for network to see both easy and hard samples during all stages of training.

Next, we are going to give the definition of sample importance in Section 2. The empirical analysis for sample importance in the deep neural network in two real datasets is discussed in Section 3. Extension about sample importance is showed in Section 4.

## 2 SAMPLE IMPORTANCE

In this section, we are going to introduce the terminology and provide a quantitative measurement of sample importance for a training procedure.

### 2.1 SAMPLE WEIGHT

In supervised learning, a model is trained by optimizing an objective over a set of observed training samples $(\mathbf{x}_i, y_i)$. Let $f(\mathbf{x}_i, \boldsymbol{\theta})$ be the output of a model for parameter $\boldsymbol{\theta}$. The training objective can be written as:

$$\sum_{i=1}^{n} L(y_i, f(\mathbf{x}_i, \boldsymbol{\theta})) + R(\boldsymbol{\theta}), \tag{1}$$

where $L(y_i, f(\mathbf{x}_i, \boldsymbol{\theta}))$ is the loss on sample $i$, and $R(\boldsymbol{\theta})$ is the regularization on the parameters. In order to highlight contribution of each sample, we can introduce sample specific weights $v_i \in [0, 1]$ which scale sample's contribution to the loss. Hence, the objective in (1) can be rewritten as:

$$\sum_{i=1}^{n} v_i L(y_i, f(\mathbf{x}_i, \boldsymbol{\theta})) + R(\boldsymbol{\theta}), \tag{2}$$

We define the weight $v_i$ as the **sample weight**. Similar definitions on $v_i$ has been proposed in Self-paced learning (SPL) Kumar et al. (2010).

In Stochastic Gradient descend (SGD) methods, parameters $\boldsymbol{\theta}$ are updated with a certain step size $\eta$ in each iteration with regard to a set of training samples. If we allow different sample weights in different iterations, a single update can be written as:

$$\boldsymbol{\theta}^{t+1} = \boldsymbol{\theta}^t - \eta \sum_{i=1}^{n} v_i^t \mathbf{g}_i^t - \eta \mathbf{r}^t,$$

where $\boldsymbol{\theta}^t$ is the parameter vector at epoch $t$, $\mathbf{g}_i^t = \frac{\partial}{\partial \boldsymbol{\theta}^t} L_i(y_i, f(\mathbf{x}_i, \boldsymbol{\theta}^t))$, $\mathbf{r}^t = \frac{\partial}{\partial \boldsymbol{\theta}^t} R(\boldsymbol{\theta}^t)$, and $v_i^t$ is the weight of $i$th sample at iteration $t$.

### 2.2 SAMPLE IMPORTANCE

If we change the weight of a sample $i$ at iteration $t$, how would such change impact the parameter training in that iteration? We can answer this question by calculating the first order derivative of parameter change $\Delta \boldsymbol{\theta}^t = \boldsymbol{\theta}^{t+1} - \boldsymbol{\theta}^t$ with regard to sample weight $v_i^t$:

$$\phi_i^t = \frac{\partial}{\partial v_i^t} \Delta \boldsymbol{\theta}^t = -\eta \mathbf{g}_i^t.$$

We call $\phi_i^t$ the **parameter affectibility** by $i$th sample at iteration $t$. $\phi_i^t$ is a vector consists of parameter affectibility from all parameters in the network. Specifically, $\phi_{i,j}^t$ is the parameter affectibility for $j$th parameter in the network. $\phi_i^t$ reflects the relationship between parameter change and different samples.

Typical deep networks contains millions of parameters. Hence, we are going to focus on groups of parameters of interests. We define $i$th **sample's importance** for parameters of $d$th layer of as:

$$\beta_{i,d}^t = \sum_{j \in \mathcal{Q}_d} (\phi_{i,j}^t)^2,$$

where $\mathcal{Q}_d$ is a set consists of the indexes of all parameters in layer $d$. Hence, **sample's importance** for all the parameters in the model is:

$$\alpha_i^t = \sum_j (\phi_{i,j}^t)^2,$$

Obviously, we have $\alpha_i^t = \sum_{d=1}^{D} \beta_{i,d}^t$.

The sum of sample's importance across all iterations is defined as overall importance of a sample:

$$\tau_i = \sum_t \alpha_i^t$$

In general, for each sample $i$, computing $\beta_{i,d}^t$ allows us to decompose its influence in the model's training across training stages and different layers.

We note that the sample importance is a high-level measurement of the samples influence on parameters at each iteration of the update. This quantity is not an accurate measurement of the relationship between a sample and final trained model. Refinements of this concept are discussed in Section 4.

## 3    EMPIRICAL ANALYSIS OF SAMPLE IMPORTANCE

We are going to explore the samples' importance for different layers at different epoch through a series of empirical experiments on two standard datasets.

### 3.1    EXPERIMENT SETUP

**Dataset**    All the analysis are performed on two standard datasets: **MNIST** [1] (LeCun et al. (1998)), a benchmark dataset that contains handwritten digit images. Each sample is a $28 \times 28$ image from 10 classes. We used 50000 samples for training and 10000 samples for testing. **CIFAR-10** [2] (Krizhevsky & Hinton (2009)), a dataset contains $32 \times 32$ tiny color images from 10 classes. Each sample has 3072 features. We used 50000 samples for training and 10000 samples for testing.

**Architecture**    We used a multilayer feed forward neural network with 3 hidden layers of 512 hidden nodes with rectified linear units (ReLU) activation function, a linear output layer, and a softmax layer on top for classification. The weights in each hidden layer are initialized according to Glorot & Bengio (2010). For hyper-parameters, we used learning rate of 0.1, batch size of 100, 50 total epochs, and weight-decay of $1e-5$. No momentum or learning decay was used. All the code are based on a common deep learning package Theano from Bergstra et al. (2010); Bastien et al. (2012).

### 3.2    SAMPLE IMPORTANCE IS STABLE WITH RESPECT TO DIFFERENT INITIALIZATIONS

Firstly, we want to explore whether the sample importance is stable under different initializations. We used three different random seeds to initialize the network parameters and calculated the sample importance every five epochs. We computed the Spearman's rank correlation between sample importance to the model, $\alpha_i^t$, in each pair of initializations. This correlation remains high in all epochs,

---

[1] http://yann.lecun.com/expdb/mnist/
[2] https://www.cs.toronto.edu/~kriz/cifar.html

above 0.9, as shown in Figure 1. This indicates that the sample importance is relatively stable to initialization of the network. Hence, all the following analysis are based on the results from initialization seed 1. (Details of training and test error for the chosen model can be viewed in Appendix Figure 9).

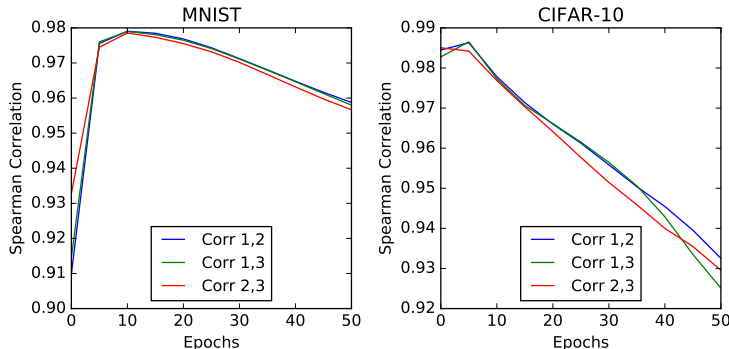

Figure 1: **Does initialization affect sample's importance?** Sample importance is preserved between initializations of the network. For each epoch, and a pair of initializations, we computed Spearman Correlation of samples' importance. Across all epochs, the correlation is of greater than 0.9 in both MNIST and CIFAR-10. Early epochs show higher consistency between ranks of sample importance across different initializations.

### 3.3 DECOMPOSITION OF SAMPLE IMPORTANCE

To better understand and visualize the sample importance, we firstly calculate the overall sample importance at each epoch as $A^t = \sum_{i=1}^{n} \alpha_i^t$. Similarly, the overall sample importance to layer $d$ is $B_d^t = \sum_{i=1}^{n} \beta_{i,d}^t$. We show the overall sample importance and its decomposition in layers for two datasets in Figure 2. Firstly, we found that even with a fixed learning rate, the overall sample importance is different under different epochs. Output layer always has the largest average sample importance per parameter, and its contribution reaches the maximum in the early training stage and then drops. Each layer contributes differently to the total sample importance. In both MNIST and CIFAR-10, the 2nd layer contributes more than the 3rd layer. In CIFAR-10, the 1st layer's provides largest contribution the total sample importance, as it contains much more parameters than other layers. Interestingly, all classes do not provide the same amount of sample importance.

We found that most samples have small sample importance (Appendix Figure 10). To visualize the contribution of different samples, we split the samples based on their total importance into three groups: 10%, top 10% – top 20% most important samples, and other 80% samples. We show the decomposition of importance contribution in each layer in Figure 3. In MNIST, the top 10% samples contribute almost all the sample importance. In CIFAR-10, most important samples contribute more in lower layers rather than output layer. This result indicates that top 20 % most important samples contribute to the majority of the sample importance.

### 3.4 SAMPLE IMPORTANCE AND NEGATIVE LOG-LIKELIHOOD

Negative log likelihood (NLL) is the loss metric we used for training objective. It has been used to measure the "easiness" of a sample in Curriculum learning Bengio et al. (2009) and Self-paced learning Kumar et al. (2010). Intuitively, the samples with large NLL should also have large sample importance (SI). However, in our experiment, we found that this is not always the case. In Figure 4, we found that 1) NLL and SI become more correlated as training goes on. However, 2) NLL is not predictive of the SI. There are many points with high NLL but small SI, and otherwise.

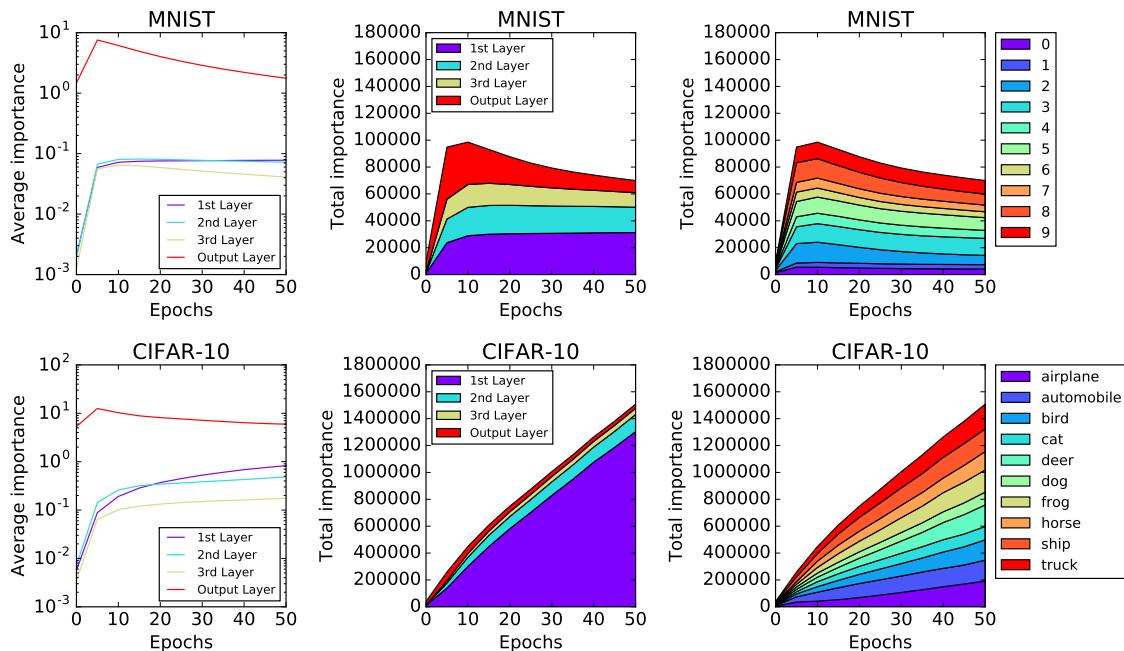

Figure 2: **Which classes and at which stage shape the network's layer's parameters?** Parameters of different layers are learned at different times. Parameters in Output layers are learned mostly during the early training stage. In the lower layers, parameters are learned predominantly during the middle and late training stage. All classes do not contribute equally to training of the model.

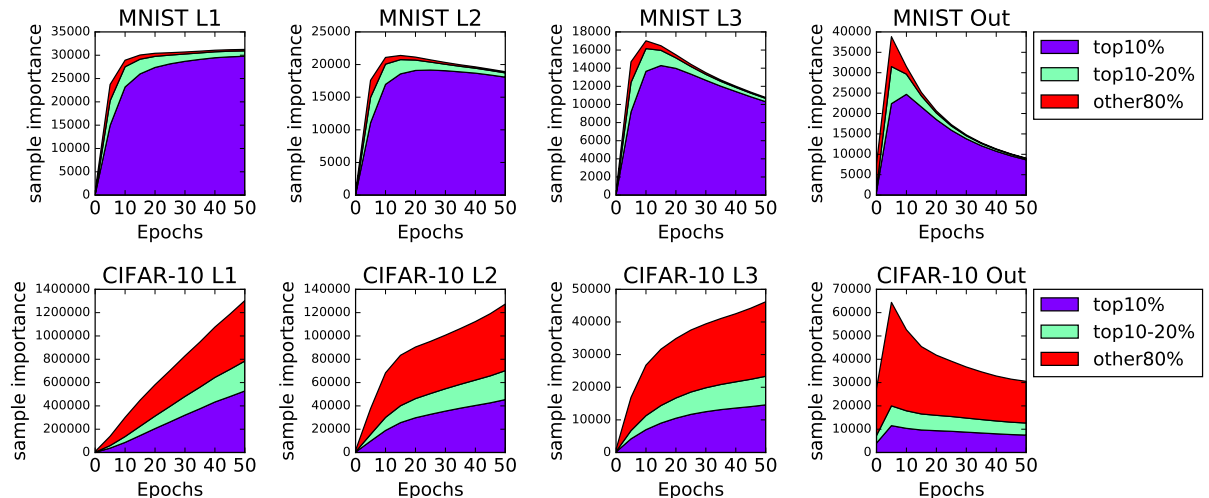

Figure 3: **Are all data samples equally important for all layers?** The top 20% most important samples contributes to the majority of parameter learning, especially in lower layers. "L1" to "L3" stands for Layer 1 to Layer 3. "Out" stands for output layer.

### 3.5 Clustering samples based on sample importance

To better visualize the importance of different samples, we provide three representative clusters of samples for each dataset. In MNIST, we clustered all digit "5" samples into 20 clusters based on their epoch-specific, layer-specific sample importance. In CIFAR-10, we clustered all "horse" samples into 30 clusters using the same features as MNIST. Kmeans algorithm is used for clustering.

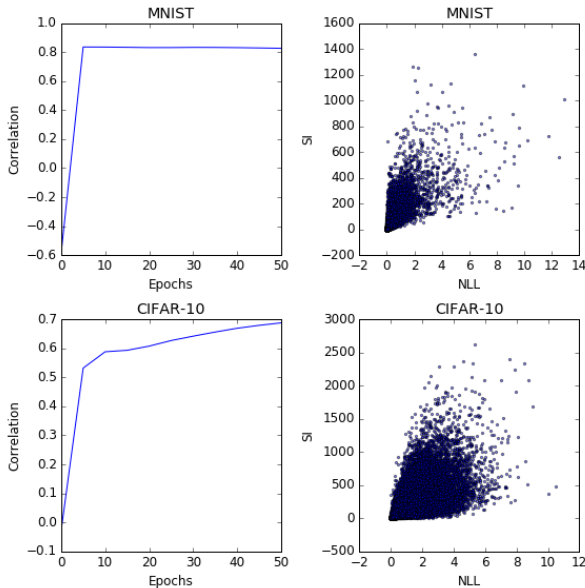

Figure 4: **Is Sample Importance correlated with Negative log-likelihood of a sample?** Sample importance is positively correlated with negative log-likelihood. As training goes on, their correlation becomes higher. However, there remain many samples with high NLL and low SI, and vice versa. Left column: correlation between sample importance and negative log likelihood for all samples across epochs. Right column: scatter plot for NLL in the last epoch and all epoch sample importance for each sample.

**MNIST**   In Figure 5, we showed 3 example clusters on digit "5". In the cluster of easy samples, where NLL converges very fast, most of the sample importance is concentrated in the first few epochs in output layer parameters. The cluster of medium samples has a slow NLL convergence compared to the easy cluster. The biggest impact is in middle training stage and more towards bottom layer parameters. Hard samples hardly converge even during the late stage of training. As training goes on, the sample importance for the bottom layer parameters become larger.

**CIFAR-10**   In Figure 6, we showed 3 examples clusters on class "horse". We observed very similar sample importance changing pattern as for the MNIST examples for easy, medium and hard clusters. Comparing to MNIST, all three clusters in CIFAR-10 have a very large impact on the parameters in the bottom layer. We note that the CIFAR-10 has almost 4 times larger number of parameters ($3075 \times 512 \approx 1574k$) in the first layer than MNIST ($785 \times 512 \approx 401k$).

## 3.6   BATCH ORDER AND SAMPLE IMPORTANCE

With the observations from empirical analysis on sample importance, we know that time – iteration – and place – layer – of sample's impact varies according to its "easiness" . We wanted to know whether constructing batches based on the sample importance or negative log likelihood would make a difference in training. Hence, we designed an experiment to explore how the different construction of batches influence the training. We note that the information used to structure the batches (negative log-likelihood and sample importance) – was obtained from a full training run.

We split all $50,000$ samples into $b = 500$ batch subsets $\{\mathcal{B}_1, \mathcal{B}_2, \ldots, \mathcal{B}_b\}$. Each batch has batch size $|\mathcal{B}_i| = 100$. In our experiment, each training sample must be in exactly one batch. There is no intersection between batches.

During training, in each epoch, we update the parameters with each batch in order $1, 2, \ldots, b$ iteratively.

We used seven different batch construction methods in this experiment:

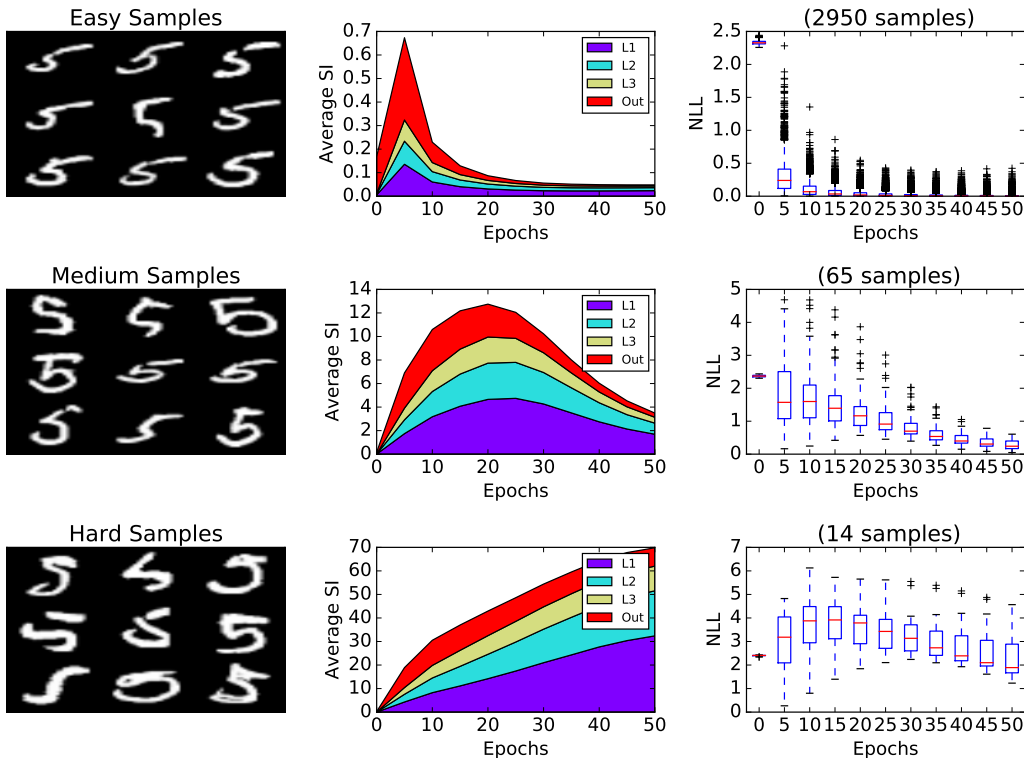

Figure 5: **When and where does an MNIST sample make the biggest impact?** For "easy" samples, their biggest impact is on output layers and during the early training stage. As sample's difficulty increases (medium and hard), the biggest impact moves to lower layers and in the late training stage. Each row is a sample cluster. In each row, from left to right: example images in the cluster; average sample importance and layer-wise decomposition across epochs; A boxplot of average training negative log likelihood across epochs.

1. **Rand**: Randomly constructed batch. All 50k samples are randomly split into $b$ batches before training. The batches and orders stay fixed during training.

2. **NLO**: Negative Log-likelihood Order. We sort all the samples based on their final NLL from low to high. The batches are constructed based on the sorted samples. First 100 samples with least NLL are in $\mathcal{B}_1$, 101 to 200 samples are in $\mathcal{B}_2$, and so on. Hence, during training, the batches with small NLL will be trained first.

3. **RNLO** Reverse-Negative Log-likelihood Order. We construct the batches same as NLO. During training, we update the batches in reverse order $\mathcal{B}_b, \mathcal{B}_{b-1}, \ldots, \mathcal{B}_1$. Hence, the batches with large NLL will be trained first.

4. **NLM** Negative Log-likelihood Mixed. We sort all the samples based on their final NLL from low to high. Next, for each sample $i$ in the sorted sequence, we put that sample into batch $j = (i \bmod b) + 1$. This ordering constructs batches out of samples with diverse levels of NLL.

5. **SIO**: Sample Importance Order. Similar to NLO, except that we sort all the samples based on their sum sample importance over all epochs from low to high. Hence, batches with small sample importance will be trained first.

6. **RSIO** Reverse-Sample Importance Order. We construct the batches same as SIO. During training, we update the batches in reverse order $\mathcal{B}_b, \mathcal{B}_{b-1}, \ldots, \mathcal{B}_1$. Hence, during training, the batches with large sample importance will be trained first.

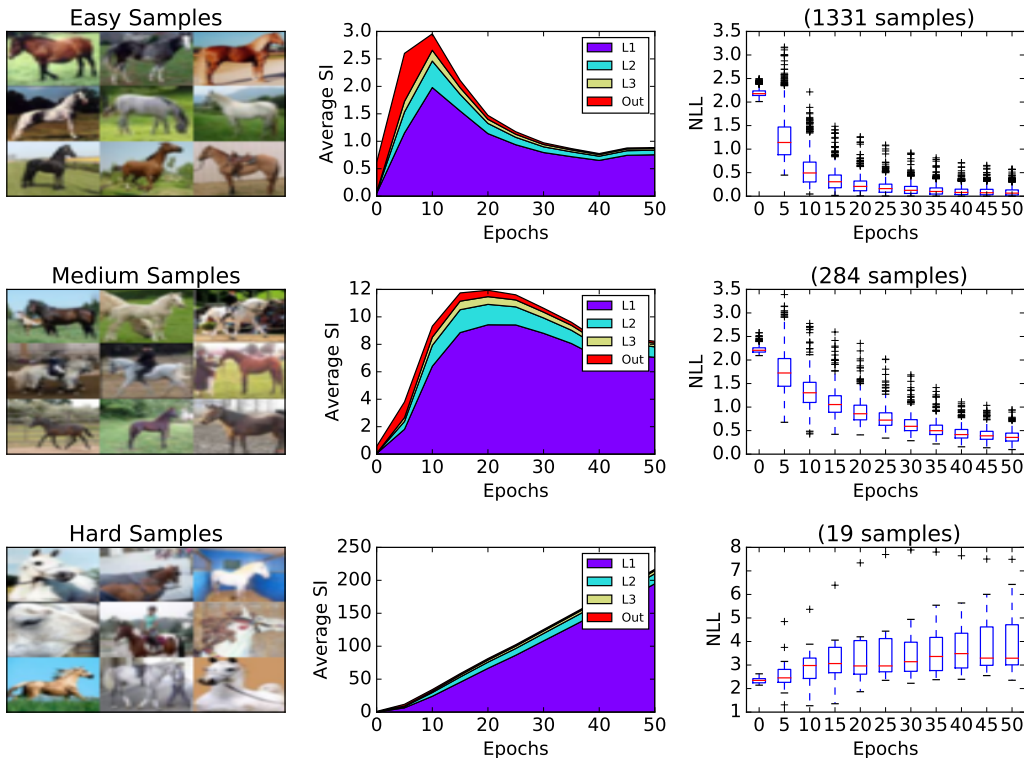

Figure 6: **When and where does a CIFAR-10 sample make the biggest impact?** For "easy" samples, their biggest impact is on the first layer during the early training stage. As samples's difficulty increases (medium and hard), the biggest impact moves to lower layers and to late training stage. Each row is a sample cluster. In each row, from left to right: example images in the cluster; average sample importance and layer-wise decomposition across epochs; A boxplot of average training negative log likelihood across epochs.

7. **SIM** Sample Important Mixed. Similar to NLM, but we sort the samples based on overall sample importance. Thus, batches contain samples with divers sample importance.

We performed five different runs (with different random initializations) on MNIST and CIFAR-10. The result is shown in Figure 7. From the result, we found that: 1) In both MNIST and CIFAR-10, Rand, SIS, and NLS have the lowest test error compared to all other methods. This indicates that diverse batches are helpful for training. 2) NLO and SIO got the worst performance in CIFAR-10. Their training error even goes up after the early stage. RNLO and RSIO have same batch constructions as NLO and SIO, but their performances are drastically different. This indicates that the order of batches during training is important. Further, training on easy samples first and hard later seems to be counter-productive.

To better understand the impact of different batch construction, we performed the principle component analysis on the learned parameters in each epoch (Figure 8). In MNIST, the impact of batch construction is not very significant. In CIFAR-10, batch construction and even the order of batch training do have a large impact on the training.

Our experiment result shows a different conclusion to Curriculum Learning and Self-paced learning, where easy samples are trained on before introducing hard samples. We found that constructing and ordering the batches – hard to easy and easy to hard – seems to hinder the performance of learning. Having hard samples mixed in with the easy ones in each batch helps the training.

Also, the results show that we want to learn from the hard samples in early epochs and "see" hard samples more frequently, even if their major impact on parameters is during the late stage. As hard

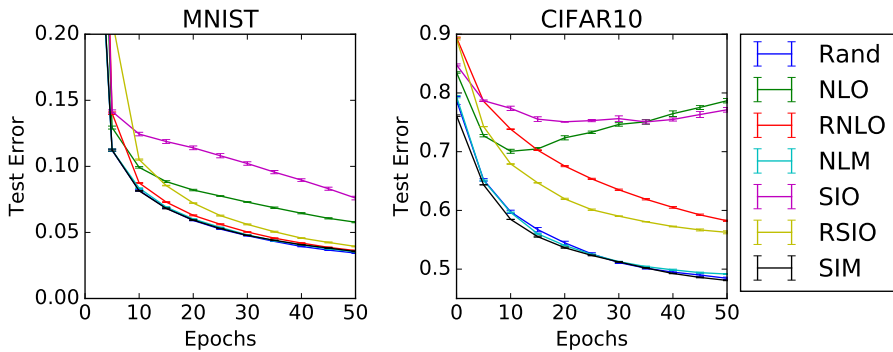

Figure 7: **Does organizing batches by "easiness" affect training?** When batches are constructed with homogeneous easiness, the training performance become worse. Batches with mixed easiness have lower test error. The solid color line represents the mean over 5 runs. The error bar indicates the standard error over 5 runs.

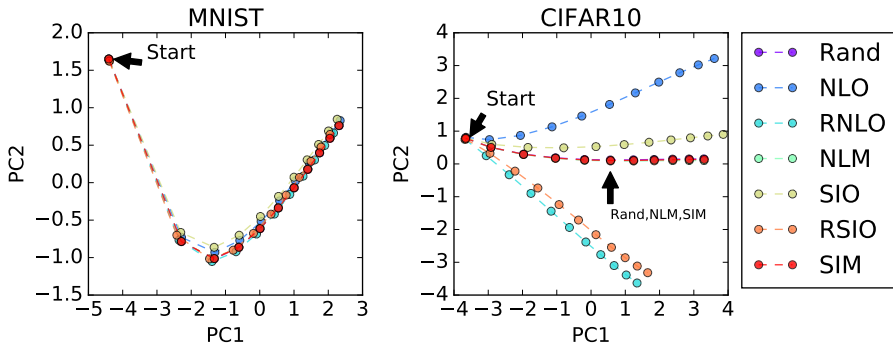

Figure 8: **Do parameters converge differently under different batch construction?** In MNIST, the converging path for all batch constructions are very similar. In CIFAR-10, batch construction with mixed easiness (Rand, NLM, SIM) has a very different converging path with all other methods. Notably, we found that even with same batch constructions but just reversed order (NLO vs. RNLO, SIO vs. RSIO), the parameters converge to different points. Each circle dotted line shows the path of the first two principle components of all parameters in different epochs. Note that in CIFAR-10, the paths of Rand, NLS and SIS are very similar and they are overlapped in the plot.

examples are few compared to easy samples and hard examples need a longer time to train, we do want to mix the hard samples into each batch to start learning from those samples early and learn longer.

## 4 EXTENSIONS OF SAMPLE IMPORTANCE

We calculated the sample importance in each iteration in Stochastic Gradient Descent. However, such quantity only reflects the impact on the change in parameters within each iteration. The influence of a sample at a particular iteration can be accumulated through updates and impact the final model. Here, we are going to derive the exact calculation of the sample's influence to the model. We rewrite the Objective (2) in Section 2 here:

$$\min_{\boldsymbol{\theta}} \sum_{i=1}^{n} v_i L(y_i, f(\mathbf{x}_i, \boldsymbol{\theta})) + R(\boldsymbol{\theta}), \tag{3}$$

Here, we deem the sample weight $v_i$ is fixed across all iterations. The update rule for stochastic gradient descent in each iteration is:

$$\boldsymbol{\theta}^{t+1} = \boldsymbol{\theta}^t - \eta \sum_i v_i \mathbf{g}_i^t - \eta \mathbf{r}_i^t$$

The derivative of $\boldsymbol{\theta}^{t+1}$ with respect to sample weight $v_i$ is:

$$\frac{\partial}{\partial v_i} \boldsymbol{\theta}^{t+1} = \frac{\partial}{\partial v_i} \boldsymbol{\theta}^t - \eta \mathbf{g}_i^t - \eta H(\boldsymbol{\theta}^t) \frac{\partial}{\partial v_i} \boldsymbol{\theta}^t$$

$$\frac{\partial}{\partial v_i} \boldsymbol{\theta}^1 = -\eta \mathbf{g}_i^0,$$

where $H(\boldsymbol{\theta}^t)$ is the Hessian matrix of the objective in (2) with regard to all parameters in iteration $t$. If we iterate the updates until convergence, then we can assume that $\boldsymbol{\theta}^T$ is a fix-point, $\boldsymbol{\theta}^* = \boldsymbol{\theta}^{T+1} = \boldsymbol{\theta}^T$, and we obtain:

$$\frac{\partial}{\partial v_i} \boldsymbol{\theta}^{T+1} - \frac{\partial}{\partial v_i} \boldsymbol{\theta}^T = -\eta \mathbf{g}_i^t - \eta H(\boldsymbol{\theta}^*) \frac{\partial}{\partial v_i} \boldsymbol{\theta}^*$$

Hence, the derivative of parameters in the final model with regard to a sample weight is:

$$\frac{\partial}{\partial v_i} \boldsymbol{\theta}^* = -H(\boldsymbol{\theta}^*)^{-1} \mathbf{g}_i^T \tag{4}$$

Equation (4) indicates that we can calculate the sample specific impact on final trained model by using the parameters learned at the convergence point. In deep learning methods, due to early stopping, fix point might not be achieved, and Equation (4) might not be an accurate.

For any target quantity $\mathcal{T}(\boldsymbol{\theta}^*)$ that depends on the final trained parameter $\boldsymbol{\theta}^*$, we can calculate the impact of a particular sample on that target as:

$$\frac{\partial}{\partial v_i} T(\boldsymbol{\theta}^*) = \frac{\partial}{\partial \boldsymbol{\theta}^*} \mathcal{T}(\boldsymbol{\theta}^*) \frac{\partial}{\partial v_i} \boldsymbol{\theta}^* \tag{5}$$

For example, if we are interested in the sum of predictions on a set of samples $\mathcal{T}(\boldsymbol{\theta}^*) = \sum_{i \in \mathcal{S}_c} f(\mathbf{x}_i, \boldsymbol{\theta}^t)$, we can use Equation (5) to calculate the derivative:

$$\frac{\partial}{\partial v_i} \mathcal{T}(\boldsymbol{\theta}^*) = \sum_{i \in \mathcal{S}_c} \frac{\partial}{\partial \boldsymbol{\theta}^*} f(\mathbf{x}_i, \boldsymbol{\theta}^t) \frac{\partial}{\partial v_i} \boldsymbol{\theta}^*$$

We note evaluating the exact impact of a sample, as shown above, is computationally cumbersome for all but the simplest models.

## 5 DISCUSSION

Samples' impact on the deep network's parameters vary across stages of training and network's layers. In our work, we found that easy samples predominantly shape parameters the top layers at the early training stages, while hard samples predominantly shape the parameters of the bottom layers at the late training stage. Our experiments show that it is important to mix hard samples into different batches rather than keep them together in the same batch and away from other examples.

There are many future extensions to the current work. Firstly, we want to expand our sample importance analysis to different deep learning structures, like Convolution Neural Network and Recurrent Neural Networks. Secondly, we want to use the sample importance as a guidance to extract a minimal subset of samples that are sufficient to achieve performance comparable to a network trained on the full dataset.

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

APPENDIX

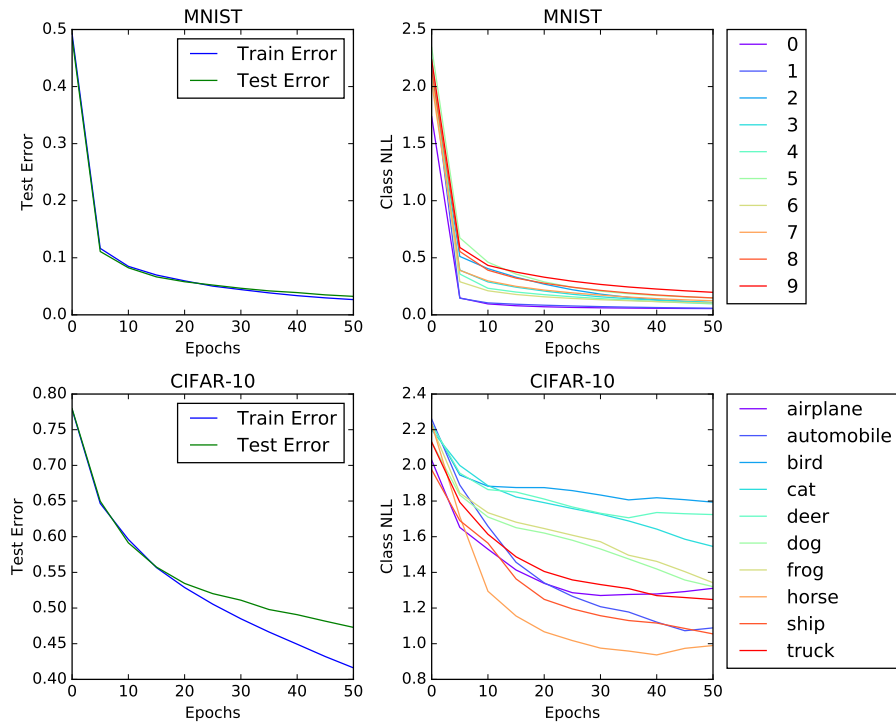

Figure 9: The training and test error on MNIST (first row) and CIFAR-10 (second row). The left column showed the average class-specific negative log likelihood.

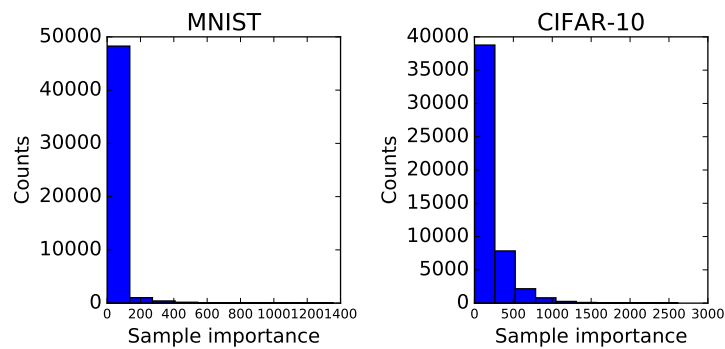

Figure 10: Histogram of total sample importance.

