# Peer review of "Sample Importance in Training Deep Neural Networks"

_ICLR 2017 — rejected_

[Public Comment · (anonymous) · 14 Dec 2016]
**Batch Selection reference is missing**

Your work is closely related to batch selection: "Online batch selection for faster training of neural networks" from ICLR Workshops 2016 where the authors provided "an initial study of possible online batch selection strategies". In your work, batch selection is denoted as "batch construction" and you study a broader set of selection strategies.

[Official Review · AnonReviewer2 · rating 3 · confidence 4 · 18 Dec 2016]
**No Title**

(paper summary) The authors introduce the notion of “sample importance”, meant to measure the influence of a particular training example on the training of a deep neural network. This quantity is closely related to the squared L2 norm of the gradient, where the summation is performed over (i) parameters of a given layer or (ii) across all parameters. Summing this quantity across time gives the “overall importance”, used to tease apart easy from hard examples. From this quantity, the authors illustrate the impact of [easy,hard] example during training, and their impact on layer depth.

(detailed review)
I have several objections to this paper. First and foremost, I am not convinced of the “sample importance” as a meaningful metric. As previously mentioned, the magnitude of gradients will change significantly during learning, and I am not sure what conclusions one can draw from \sum_t g_i^t vs \sum_t g_j^t. For example, gradients tend to have higher norms early in training than at convergence, in which case weighting each gradient equally seems problematic. I tried illustrating the above with a small thought experiment during the question period: “if” the learning rate were too high, training may not even converge in which case sample importance would be ill-defined.  Having a measure which depends on the learning rate seems problematic to me, as does the use of the L2 norm. The “input Fisher” norm, \mathbb{E} \frac{\partial \log p} {\partial x} (for a given time-step) may be better suited, as it speaks directly to the sensitivity of the classifier to the input x (and is insensitive to changes in the mean gradient norm). But again summing Fisher norms across time may not be meaningful.

The experimental analysis also seems problematic. The authors claim from Fig. 2 that output layers are primarily learnt in the early stage of training. However, this is definitely not the case for CIFAR-10 and is debatable for MNIST: sample importance remains high for all layers during training, despite a small spike early on the output layer. Fig 2. (lower, middle) and Fig. 6 also seems to highlight an issue with the SI measure: the SI is dominated by the input layer which has the most parameters, and can thus more readily impact the gradient norm. Different model architectures may have yielded different conclusions. Had the authors managed to use the SI to craft a better curriculum, this would have given significant weight to the measure. Unfortunately, these results are negative.

PROS:
+ extensive experiments

CONS:
- sample importance is a heuristic, not entirely well justified
- SI yields limited insight into training of neural nets
- SI does not inform curriculum learning

[Official Review · AnonReviewer1 · rating 7 · confidence 4 · 19 Dec 2016]
**This paper examines how the gradient of each individual data sample influences the various aspect of SGD learning for neural networks.**

This paper examines the so called "Sample Importance" of each sample of a training data set, and its effect to the overall learning process.

The paper shows empirical results that shows different training cases induces bigger gradients at different stages of learning and different layers.
The paper shows some interesting results contrary to the common curriculum learning ideas of using easy training samples first. However, it is unclear how one should define "Easy" training cases.

In addition, the experiments demonstrating ordering either NLL or SI is worse than mixed or random batch construction to be insightful.

Possible Improvements:
It would be nice to factor out the magnitudes of the gradients to the contribution of "sample importance". Higher gradient (as a function of a particular weight vector) can be affected weight/initialization, thereby introducing noise to the model.

It would also be interesting if improvements based on "Sample importance" could be made to the batch selection algorithm to beat the baseline of random batch selection.


Overall this paper is a good paper with various experiments examining how various samples in SGD influences the various aspect of training.

[Official Review · AnonReviewer3 · rating 2 · confidence 4 · 22 Dec 2016 (modified: 21 Jan 2017)]
**a few interesting ideas but very big issues**

The paper proposes a new criterion (sample importance) to study the impact of samples during the training of deep neural networks. This criterion is not clearly defined (the term \phi^t_{i,j} is never defined, only \phi^t_i is defined; Despite the unclear definition, it is understood that sample importance is the squared l2 norm of the gradient for a sample i and at time t strangely scaled by the squared learning rate (the learning rate should have nothing to do with the importance of a sample in this context).

The paper presents experiments on the well known MNIST and CIFAR datasets with correspondingly appropriate network architectures and choice of hyper-parameters and initialisations. The size of the hidden layers is a bit small for Mnist and very small for CIFAR (this could explain the very poor performance in figure 6: 50% error on CIFAR)

The study of the evolution of sample importance during training depending on layers seems to lead to trivial conclusions
 - “the overall sample importance is different under different epochs” => yes the norm of the gradient is expected to vary
 - “Output layer always has the largest average sample importance per parameter, and its contribution reaches the maximum in the early training stage and then drops” => 1. yes since the gradient flows backwards, the gradient is expected to be stronger for the output layer and it is expected to become more diffuse as it propagates to lower layers which are not stable. As learning progresses one would expect the output layer to have progressively smaller gradients. 2. the norm of the gradient depends on the scaling of the variables

The question of Figure 4 is absurd “Is Sample Importance the same as Negative log-likelihood of a sample?”. Of course not.

The results are very bad on CIFAR which discredits the applicability of those results.

On Mnist performance is not readable (figure 7): Error rate should only be presented between 0 and 10 or 20%

Despite these important issues (there are others), the paper manages to raises some interesting things: the so-called easy samples and hard samples do seem to correspond (although the study is very preliminary in this regard) to what would intuitively be considered easy (the most representative/canonical samples) and hard (edge cases) samples. Also the experiments are very well presented.

[Final Decision · Program Chairs · 06 Feb 2017]
**ICLR committee final decision**

The reviewers provided detailed, confident reviews and there was significant discussion between the parties. 
 
 Reviewer 2 and 3 felt quite strongly that the paper was a clear reject. Reviewer 1 thought the paper should be accepted.
 
 I was concerned with the two points raised by R3 and don't feel they were adequately addressed by the author's comments:
 
 - Dependence of the criteria on the learning rate (this does not make sense to me); and
 - Really really poor results on CIFAR-10 (and this is not being too picky, like asking them to be state-of-the-art; they are just way off)
 
 I engaged R1 to see how they felt about this. In reflection, I side with the majority opinion that the paper needs re-work to meet the ICLR acceptance bar.